# Less Known Is More Feared—A Survey of Children’s Knowledge of and Attitudes towards Honeybees

**DOI:** 10.3390/insects15050368

**Published:** 2024-05-18

**Authors:** Emmanuele Leto, Francesco Pace, Giulia Sciotto, Barbara Manachini

**Affiliations:** 1Department of Agricultural, Food and Forest Sciences (SAAF), University of Palermo, 90128 Palermo, Italy; emmanueleleto@gmail.com; 2Department in Economics, Business and Statistics (DSEAS), University of Palermo, 90128 Palermo, Italy; francesco.pace@unipa.it; 3Department of Psychology, Educational Sciences and Human Movement, University of Palermo, 90128 Palermo, Italy; giulia.sciotto@unipa.it

**Keywords:** *Apis mellifera*, biodiversity, perception, attitude, knowledge

## Abstract

**Simple Summary:**

Children can play a key role in biodiversity conservation as they pass knowledge on to their families who, in turn, can further disseminate it, and as children will be future citizens. This research focused on the relationship between the knowledge and perception of honeybees, which provide essential ecosystem services, in 12–14-year-old children. A survey was conducted with 231 students in Palermo (Sicily, Italy) for which they were given a questionnaire to assess their perception and knowledge of honeybees. The results indicate that the students have a good understanding of the role of honeybees in nature but lack a clear idea of their interactions with the environment. The study also reveals that children feel a certain fear of honeybees but still have respect for them. Interestingly, the average score for ecological knowledge surpassed the average score for perception, indicating that the subjects had a more positive education about honeybees than actual perception.

**Abstract:**

The global decline in the number of pollinators has elicited considerable public attention. To the general public, honeybees are considered to be the primary pollinators. Also, a decline in managed honeybee stocks is alarming and could lead to declining pollination services and reduced ecosystem biodiversity, although the *Apis mellifera* is the least likely pollinator species on the planet to be at risk of extinction. A less-than-complete understanding of honeybees and their ecology may hinder their conservation. Ascertaining the public’s level of knowledge about, and perception of, a problem can help in solving it. This research focused mainly on honeybees because people are unlikely to be able to recognize the different species of Apoidea. Schools are ideal places for understanding the basic knowledge and attitudes regarding this insect. We aimed to understand the perception and knowledge of 12–14-year-old children towards honeybees as well as to verify the existence of a correlation between knowledge level and positive perception. Secondary school students can play a key role in the conservation of biodiversity as they are carriers of knowledge in families and will be future citizens. To this end, 231 students were given a 26-item questionnaire related to their perception and knowledge of honeybees. Results indicate that the students have a good understanding of the role that bees play in nature but do not have a completely clear idea of this insect’s interactions with the environment. Results also show that the children feel a certain fear of honeybees, although they respect them. The average score of the ecological branch test exceeded the average score of the perceptual one, indicating that the subjects had a more positive education than perception.

## 1. Introduction

Pollination by insects is an ecosystem service of great value for the environment and humanity [1]. In recent years, a loss of biodiversity has been highlighted, and one of the most detrimental effects is the decline in pollinating insects [2,3]. Reduced pollination by honeybees and other pollinators can result in many detrimental consequences, such as a decline in plant species diversity and an alteration of the composition of plant communities [4].

Today, there is a global decline in the number of both non-honeybee pollinators, such as wild bees, butterflies, moths, flies, hoverflies, beetles, wasps, birds, small mammals, and in the most known pollinators, honeybees [5]. This decline has serious consequences for ecosystems, human health, and economics, reducing, for example, agricultural production, varietal diversity, and food security [6,7,8,9]. However, despite their importance and global proclamations of concern, the declining trends in honeybees and other pollinators are still ongoing [10,11,12]. *Apis mellifera* L. (Hymenoptera: Apoidea) honeybees are the most well-known pollinators to the general public.

*A. mellifera* is used by humans for the pollination of more than 150 cultivated botanical species; it also contributes to the protection, conservation, and restoration of habitats through its pollination of about 75–80% of the higher flowering plants, both wild and cultivated [13,14]. Considering only the contribution of honeybees to food production, the economic value of pollination has been estimated at between 235 and 577 billion dollars per year [15]. For example, bees support almond ($2.8 billion) and apple ($2 billion) production and many other cultivated species [16]. Honeybees have also provided us with products such as honey and wax since ancient times [17].

Italy is fourth in Europe for the number of honeybee colonies and fifth for honey production. The number of beekeepers is constantly growing; in Italy, there were over 72 thousand in 2022, 54% more than in 2017. In 2022, there were over 1.57 million (+8% vs. 2021 and +29% compared to 2017) hives present across the national territory, of which 79% were for commercial use. National honey production is estimated at approximately 23,000 tons [18]. Per capita honey consumption in the same year was approximately 450–500 g, including honey imported from abroad (28,144 metric tons for a value of over €56 million) [19]. Sicily is quite important to Italian honey production; in fact, it is the third region for the number of hives, about 106,000, with a production of about 22 kg of honey per year per hive unit [20]. Given the high production and consumption of honeybee products in Italy and Sicily, and the high biodiversity of Sicilian flora and fauna [21], we believe it is essential to understand the local population’s perception of honeybees.

Despite their great economic and ecological contributions, insects are associated with feelings of fear and disgust [22,23], which can decrease people’s interest in these animals and consequently hinder successful environmental education [24]. It is assumed that negative attitudes towards animals are due to a biological predisposition that alerts us to potentially dangerous species [25], and animals that people fear tend not to receive adequate support for their conservation [26]. Therefore, the public’s attitude towards an animal influences people’s willingness to protect it [27]. Honeybees, in particular, are associated with a feeling of danger [28].

The conservation and protection of natural resources require the involvement of society and citizens [29]. Any conservation plan needs not only the scientific community’s support but also the public’s [30]. To safeguard and protect endangered species, people need to be aware of the benefits that a species brings and how to protect it [31]. Nature conservation often depends on the behavior of individuals, which can be driven by socio-psychological factors such as a person’s attitude, knowledge, and identity [32].

It is well known that knowledge of ecological concepts and attitudes towards animals are the basis of successful conservation efforts, and ecological knowledge is the foundation of environmental education for children [33]. Thus, the more people are aware of the importance of honeybees and the pivotal role they play, the greater the personal and collective respect towards them that can be expected.

But how well are the biology of honeybees and their ecology known? How much are they talked about in schools? How well do children understand the role of this insect, and do they, indeed, really know much about it? Surprisingly, there are very few studies on the perceptions, attitudes, and knowledge related to bees, though pollinator conservation does seem to be a focus of conservation programs [34]. It is necessary to start with the schools and provide children with a clear ecological picture focusing on the importance of biodiversity as well as the conservation of these fundamental pollinators. People’s indifference towards the problems of at-risk animals can be countered by knowledge through education [35].

In a constructivist perspective, children’s conceptions and perceptions of a topic will influence their interpretation of the world [36]. Knowledge of animals and the ecosystem services they perform is thus considered the fundamental basis for promoting environmental education in children [37,38]. It has been observed that children who have inaccurate conceptions or information about an animal are more likely to demonstrate negative perceptions towards that animal [39]. Indeed, negative perceptions towards animals are often accompanied by false myths and superstitions [40], as well as other cultural elements [39,40]. Conversely, a better understanding of environmental issues is associated with more positive attitudes [41].

In this study, the aim was to evaluate students’ understanding of ecological concepts and their knowledge about honeybees, as well as to collect data on their perceptions and beliefs about honeybees. The study also aimed to examine any correlations between students’ conceptual knowledge and perceptions. It is essential to provide children with a clear ecological picture and emphasize the importance of biodiversity and the conservation of pollinators in schools. Education and knowledge can counter people’s indifference towards at-risk animals and promote positive attitudes towards honeybees and other pollinators.

## 2. Materials and Methods

### 2.1. Sample

Four lower secondary schools in the city of Palermo (Sicily, Italy) were randomly selected, and for each school, three eighth grade classes (corresponding to middle school Italian level III) were randomly chosen. The number of students in the lower secondary schools of Palermo were 21,170 (data available on https://www.comune.palermo.it/palermo-staticstica, accessed on 14 February 2024), of which the students in grade 8 were about 6900; therefore, the sample size was determined as a minimum of 225 questionnaires, given that the confidence level α = 0.95 and the measurement error is 5%. In this study, 231 questionnaires were administered to the volunteer students. The consent of the schools’ principals was obtained for the administration of an anonymous questionnaire designed to test the participants’ attitudes towards honeybees and knowledge of honeybee ecology.

In this research, we considered mainly honeybees, though we are aware that most people are unlikely to be able to recognize the different species of Apoidea and generally call them all honeybees or bees, without distinguishing among the species and sometimes even confusing them with wasps. All families were informed about the potential anonymous volunteer questionnaire through a circular letter from the school. The one-hour questionnaire was filled in manually by the volunteer students under the supervision of their teachers and an assistant from the University of Palermo. The students and their families were notified by the schools in writing. Even though it would have been simpler and quicker, the questionnaire was not administered online to avoid the possibility that students might be tempted to check the answers on the Internet, falsifying the real results regarding their knowledge and/or attitudes. There was also close surveillance to ensure that the responses were individual. All information acquired during the study was kept anonymous to protect the participants’ privacy.

### 2.2. The Questionnaire

A questionnaire consisting of 26 statements was developed to assess the students’ knowledge of bee ecology and their attitudes towards bees; 18 items concerned knowledge about bees and 8 items were about perceptions and attitudes towards them. To measure the reliability of the scales, Cronbach’s alpha coefficient was calculated [42]. Both scales were found to be reliable, with Cronbach’s alpha values of 0.65 and 0.73, respectively [43].

Each item was rated on a 5-point Likert scale [44]. Subjects were also asked to indicate their gender, place of residence (countryside or city), and whether they liked to observe nature and spend time outdoors.

### 2.3. Scoring and Data Analysis

Scores given to correct statements regarding bee knowledge were as follows: 5 points for “Strongly agree”, 4 points for “Agree”, 3 points for “Don’t know”, 2 points for “Disagree”, and 1 point for “Strongly disagree.” Scores given to incorrect statements were as follows: 1 point for “Strongly agree”, 2 points for “Agree”, 3 points for “Don’t know”, 4 points for “Disagree”, and 5 points for “Strongly disagree.” In the attitude scale, 5 stands for “Strongly agree”, 4 for “Agree”, 3 for “Don’t know”, 2 for “Disagree”, and 1 for “Strongly disagree”, such that higher scores indicated annoyance by, or a general attribution of negative characteristics to, bees. For the knowledge scale, based on statements aimed at exploring how informed students were about bees and their behavior in the ecosystem, each question gave rise to a score, depending on whether the subject agreed or disagreed with a truthful or incorrect statement.

The data were subsequently subjected to statistical analysis using Student’s t-index and one-way ANOVA to verify the differences between the means according to the subjects’ gender, residence in the city or countryside, and their fondness for nature and the outdoors. Correlational analyses were also conducted between the calculated indicators.

## 3. Results

### 3.1. Sample

All the 231 questionnaires administered were fully completed. The average age of the participants was 13 years old, 54.5% were male and 45.5% female. Children living in the city were 78.8% of the sample, 19% were living in the countryside, 1.3% stated they had the option of living in both, and 0.9% did not answer. When asked if they spent a lot of time outdoors, 76.6% of the participants gave a positive response and 23.4% a negative one. Regarding the item on nature observation, 97.4% of the sample responded affirmatively, and only 2.6% answered negatively.

### 3.2. Responses to the Questionnaire

#### 3.2.1. Knowledge of Bees

The results of the frequency analyses relating to bee knowledge are reported in Table 1. Figure 1 shows the distribution of the knowledge scale scores in the secondary school students aged 12–14 years.

The level of knowledge was divided into four clusters, based on percentiles, to divide the students into four groups. Overall, 30.1% show a low level of knowledge, 20.4% a medium–low level, 29.6% a medium–high level, and 19.9% a high level of knowledge. We tried to calculate the chi-square to test whether the level of knowledge depended in any way on gender (χ = 3.891, df = 3, *p* value = 0.273), city or countryside residence (χ = 3.880, df = 6, *p* value = 0.693), or spending a lot of time outdoors (χ = 3.191, df = 3, *p* value = 0.363). The test was not statistically significant, so it can be concluded that there is no relationship between gender, place of residence, and propensity to spend leisure time outdoors and knowledge about honeybees.

The average value of the scores relating to knowledge of honeybee ecology (item C12) was 3.27, with a score < 4.00 denoting poor knowledge of honeybees and potentially other bees. However, the subjects demonstrated knowledge of the importance of honeybees as pollinators, obtaining an average score of 4.26 on item C13, and the cognition that bees sting to defend themselves, obtaining an average score of 4.50 on item C6, the highest score in the ecological section.

No significant differences were found in knowledge between subjects coming from the countryside and those coming from the city (t = 0.381, df = 219, *p* = 0.352). However, the children living in the countryside showed a greater knowledge of the role of honeybees as pollinators (item C13: t = −2.141, df = 224, *p* = 0.017), scoring an average score of 4.50, while those living in the city scored 4.20. In general, knowledge of honeybees and their role was higher among males (t = 1.698, df = 223, *p* = 0.045).

#### 3.2.2. Attitudes towards Bees

The results of the frequency analyses relating to attitudes towards honeybees are reported in Table 2, while Figure 2 shows the distribution of the attitude scale scores.

Attitude levels were divided into four clusters, based on percentiles, to subdivide the students into four groups.

Overall, 20.8% showed quite negative attitudes towards honeybees, 28.1% a medium–low level perception, 25.8% a medium–high level perception, and 25.3% of the students showed decidedly positive attitudes towards honeybees. The average value of the scores relating to attitudes towards honeybees was 2.40, with a score > 4.00 denoting an overall non-positive attitude of the children. Also, in this case, we calculated the chi-square to test whether the negative attitude towards honeybees depended in any way on gender (χ = 2.576, df = 3, *p* value = 0.462), city or countryside residence (χ = 6.329, df = 6, *p* value = 0.387), or spending a lot of time outdoors (χ = 6.097, df = 3, *p* value = 0.107). The test was not statistically significant, so it can be concluded that there is no relationship between gender, place of residence, and propensity to spend leisure time outdoors and the negative attitude towards honeybees.

No significant differences were found between males and females (t = −1.17, df = 218, *p* = 0.122) regarding their negative attitude towards honeybees. However, some statistically significant differences between means emerged in the responses to items P4, P7, and P8. Items P4 and P7 indicated, respectively, feeling threatened by the presence of honeybees and the greater perceived enjoyment of spending time outdoors if there were fewer honeybees, while item P8 indicated the inclination to grant permission to build hives near one’s home. Overall, males were less disturbed by honeybees than females (t = −3.12, df = 227, *p* = 0.001 for item P4; t = −1.99, df = 223, *p* = 0.024 for item P7) and more inclined to favor the presence of hives near their home (t = 2.11, df = 226, *p* = 0.018). Statistically significant differences were also found in the attitudes towards honeybees between children from the countryside and those from the city (t = 1.48, df = 216, *p* = 0.032).

Finally, the *t*-test used to verify the differences between the means of the scores on the negative attitudes scale based on the subjects’ preference for spending their free time in the open air revealed a statistically significant result. Those who indicated that they did not spend a lot of time outdoors showed higher average scores, therefore showing more negative attitudes towards honeybees compared to those who stated they spent a lot of time outdoors (t = 2.74, df = 217, *p* = 0.003). On the contrary, those who did not spend much time outdoors felt much more threatened by honeybees (t = 2.72, df = 227, *p* = 0.004 for item P4).

### 3.3. Correlations between Knowledge and Attitude

The results showed a significant negative correlation between the level of knowledge and a negative attitude towards honeybees (Pearson’s correlation = −0.430, *p* < 0.001), so levels of knowledge about honeybees and negative attitudes towards them were inversely proportional. Therefore, the feelings of being threatened and attitudes of rejection towards the presence of honeybees could be linked to a lack of knowledge about them (see Figure 3).

## 4. Discussion

Observation is the fundamental basis of all research, and young people must spend time observing the world around them, asking questions, and reflecting on the how and why of the fascinating processes that nature offers us; in addition, the observation of and contact with nature help increase cognitive flexibility [41].

It is evident from the answer to the question, “Do you like observing nature?” that the children of Palermo not only like to spend time outdoors but also to stop and observe their surroundings. Spending a lot of time outdoors can increase the likelihood that children observe and come into contact with animals such as honeybees or other bees, which are usually rarer in urban habitats, thus leading to better knowledge of these creatures [45]. Indeed, such knowledge promotes conservation activities as people rarely protect and appreciate things that they do not know well [46].

The results related to the statements about the feeding activity of honeybees indicate that young people do not have very clear ideas about the morphology and diet of this insect, and therefore they can also confuse honeybees with wild bees. Knowing an animal’s basic anatomy and having direct experiences with it can help in evaluating the risks that its proximity can bring [47]. As can be seen from the answers, these young people were unaware that honeybees cannot bite because they do not have the necessary mouthparts. In this research, we assumed that the children were not aware of a particular behavior that honeybees adopt towards invertebrate invaders of the hives. Indeed, studies reported that honeybees use their mandibles to bite invaders that are too small to sting [48]. Hive intruders, such as wax moth larvae *Galleria mellonella* L. (Lepidoptera: Pyralidae) and the parasitic mite *Varroa destructor* (Anderson and Trueman; Mesostigmata: Varroidae), can be paralyzed for a short time after being bitten by honeybees. Invaders are probably anaesthetized by specific compounds secreted from the mandibular glands during biting [48]. However, we believe that the knowledge of this very interesting but specific behavior of honeybees is not known by most people and is limited primarily to researchers and/or professionals.

Another interesting result is the score of 3.70 for statement C15. From this score, it seems that the concept of biodiversity is not yet completely clear to the middle school students. Having a clear understanding of the concept of biodiversity is a key point for the proper protection of any living being, and the result of this question demonstrates that we need to focus more attention on this issue.

The protection and conservation of insects does not happen by chance; not only must researchers and scholars know the biology of these insects, but people in general as well [49,50], especially children who, in the future, will hopefully be active in the fight to protect them [51].

As suggested by other studies regarding the protection of pollinating insects [10,52], as well as from the results of our study, it is of fundamental importance to develop educational materials that can increase middle school students’ knowledge about honeybees and the concept of biodiversity, thus enabling them to actively participate in protection programs and, above all, making them more aware that this insect is not a threat but rather an invaluable asset for humanity. Our research was limited to the study of honeybees and other bees which are certainly the best known to children as well as the majority of the population. However, further investigations would be interesting to understand both the knowledge and the attitude towards wild bees which have a very important role both in pollination and in the maintenance of biodiversity in the natural and urban environments, along with the importance of urban gardens in the conservation of solitary bees, which has recently been highlighted [53].

### Limitations

This study has several limitations. Firstly, the participants in this study are only from Palermo middle schools; therefore, the results might not be an accurate representation of young students in another contexts. The second limitation is that the participants’ responses to the questionnaire may have been influenced by their close associates. Since purposive and snowballing sampling were used, there is a possibility of bias because an equal number of students from each school were not selected. Studies in the future could focus on a wider range of students to increase the validity of the results. Lastly, the study focused on honeybees. Recent studies confirm that people are more familiar with *A. mellifera* compared to non–*Apis* bee species [54]. However, no other bees or similar taxa are specifically considered in our study, and this might have had an influence on the results. Awareness of the decline in local honeybees has raised the profile of pollinators in general, but the disproportionate level of public attention to honeybees now needs to be extended to a wider range of pollinators, especially those who are threatened or in grave danger [55]. However, these findings highlight the importance of knowledge to encourage the conservation and protection of honeybees and other pollinators. Nevertheless, while interest in honeybees and other insect pollinators has grown over the past few decades, largely as a result of their economic importance, a general understanding of the importance of pollinator diversity and ecology, as well as the factors contributing to the decline in some species, has not yet been achieved [55].

## 5. Conclusions

This study allowed us to assess middle school students’ perception and knowledge regarding honeybees and their ecological importance. The results demonstrated a good knowledge on the part of the participants concerning the importance of honeybees, but some lack of awareness of their biology and ecology. This lack of knowledge was mostly associated with the lack of spending time outdoors, living in the city. So, the next step would be to understand why urban students perceive and define bees differently. It is also interesting to note that students who say they spend less time outdoors feel more afraid of bees, highlighting how the lack of contact with nature alters their perception. It would be necessary to determine if the current generation of young people reap the suggested benefits of that connection or are exposed to the consequences of insufficient contact with honeybees and nature in general.

On the other hand, all students demonstrated a respect for this insect and the need to protect it. It is noteworthy that generating broader knowledge and positive attitudes towards honeybees and other important insect taxa is a means of promoting positive behavioral changes surrounding the reduction in biodiversity. Our results reinforce what some authors have asserted, that it is essential to be able to exploit the popularity of honeybees as a conservation tool through better education, clear public messages, and science communication [55].

## Figures and Tables

**Figure 1 insects-15-00368-f001:**
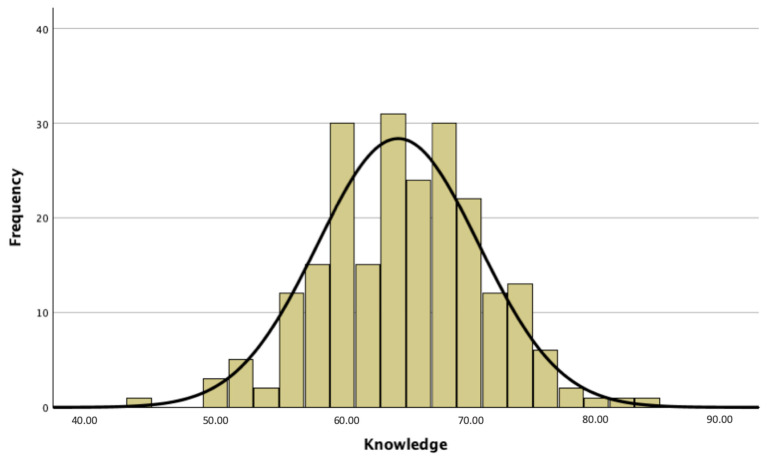
Distribution of the knowledge scale scores in the 12–14-year-old secondary school students.

**Figure 2 insects-15-00368-f002:**
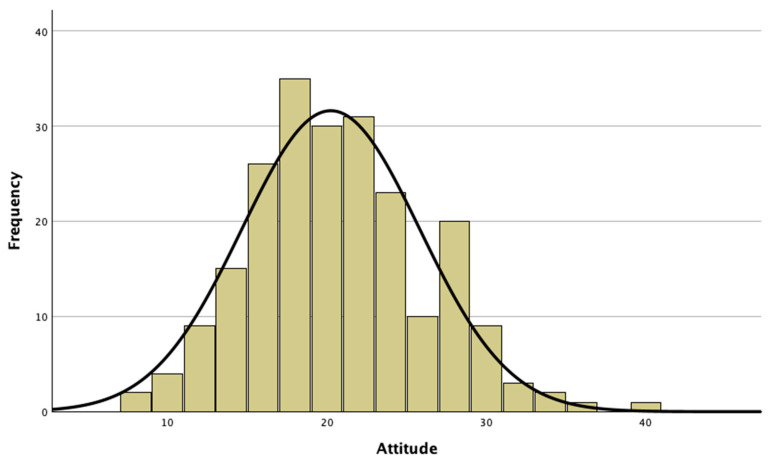
Distribution of attitude scale scores in the 12–14-year-old secondary school students.

**Figure 3 insects-15-00368-f003:**
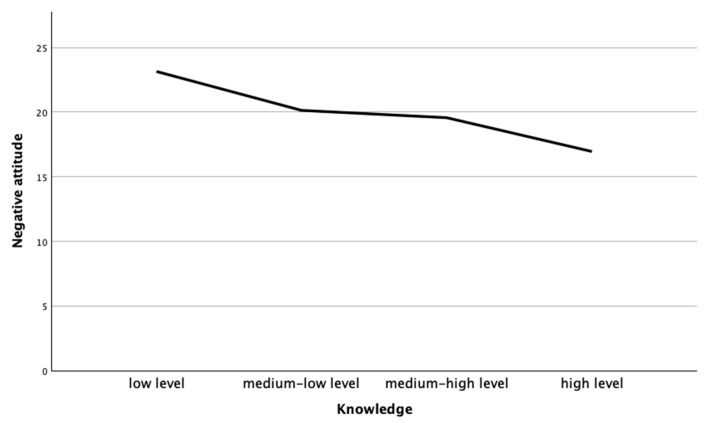
Correlations between level of knowledge about honeybees and attitude in 12–14-year-old secondary school students.

**Table 1 insects-15-00368-t001:** Frequency of response for each item about bee-related knowledge and ecological concepts.

Knowledge and Ecological Concepts	% Frequency of Responses in Likert Scale					Adjusted Mean Score (SD)				
	SA	A	NE	D	SD	Total	Boys	Girls	City	Countryside
(C1) Honeybees bite	6.1	10	13	24.2	46.8	2.04 (1.24)	1.98 (1.26)	2.11 (1.21)	2.00 (1.23)	2.20 (1.30)
(C2) Honeybees feed on pollen	44.2	33.8	12.6	6.9	2.6	4.1 (1.04)	4.11 (1.09)	4.08 (0.98)	4.07 (1.05)	4.16 (1.03)
(C3) Honeybees eat fruit	2.6	9.1	50.6	16.9	20.8	2.56 (1)	2.45 (1.02)	2.70 (0.97)	2.58 (1.00)	2.48 (1.02)
(C4) Honeybees sting in order to suck blood	3	4.3	5.2	23.4	64.1	1.59 (0.99)	1.46 (0.93)	1.74 (1.04)	1.58 (1.00)	1.66 (0.96)
(C5) Honeybees can use nectar or hunt other life forms for food	13.4	35.1	36.8	11.3	3.5	3.44 (0.97)	3.49 (0.97)	3.38 (0.98)	3.44 (0.97)	3.45 (0.97)
(C6) Honeybees sting to defend themselves	64.3	26.5	5.2	2.6	1.3	4.5 (0.82)	4.54 (0.82)	4.45 (0.82)	4.5 (0.82)	4.5 (0.85)
(C7) Honeybees like to sting	2.2	3.5	23.4	28.6	42.4	1.94 (0.99)	1.85 (0.96)	2.03 (1.03)	1.92 (1.03)	2.02 (0.88)
(C8) Honeybees mainly sting humans	3	29.9	26.8	30.3	10	2.86 (1.05)	2.72 (1.07)	3.01 (1.01)	2.84 (1.04)	2.91 (1.14)
(C9) People are often stung by honeybees	9.1	48.7	22.6	15.7	3.9	3.43 (0.99)	3.37 (1.02)	3.5 (0.95)	3.43 (0.99)	3.48 (1.04)
(C10) When you have a picnic, honeybees approach the food to eat it	19.6	37.8	23.5	12.2	7.0	3.51 (1.14)	3.55 (1.17)	3.47 (1.12)	3.51 (1.15)	3.5 (1.11)
(C11) Fewer honeybees in the world would benefit agriculture	8.7	15.7	18.8	19.7	37.1	3.501.3)	2.23 (1.35)	2.56 (1.32)	2.35 (1.36)	2.59 (1.32)
(C12) Pollution is responsible for the decline in the honeybee population	13.9	27.7	39.4	10	9.1	3.27 (1.11)	3.34 (1.12)	3.21 (1.1)	3.23 (1.15)	3.5 (0.93)
(C13) Honeybees play the role of pollinators in our ecosystems	46.3	36.4	15.6	0.4	1.3	4.26 (0.83)	4.34 (0.88)	4.17 (0.76)	4.20 (0.87) *	4.50 (0.59) *
(C14) The balance of our ecosystems would fail if there were no more honeybees	32.5	30.7	24.7	8.2	3.9	3.80 (1.1)	3.87 (1.08)	3.7 (1.13)	3.91 (1.04)	3.43 (1.21)
(C15) A decline in the bee population would not affect ecosystems	7.4	6.6	21	19.2	45.9	3.70 (1.2)	2.07 (1.32)	2.14 (1.2)	2.06 (1.24)	2.25 (1.31)
(C16) Habitat reduction is responsible for the decline in the honeybee population	15.7	29.1	45.2	6.5	3.5	3.47 (0.95)	3.55 (1)	3.37 (0.89)	3.46 (0.93)	3.48 (1.05)
(C17) A decline in the honeybee population would increase the biodiversity of our ecosystems	6.1	13.9	52.8	11.3	16.0	2.83 (1.05)	2.85 (1.07)	2.8 (1.05)	2.79 (1.04)	2.89 (1.06)
(C18) The uncontrolled use of insecticides is responsible for the decline in the honeybee population	37.7	37.7	18.2	3.5	3.0	4.03 (0.99)	4 (1.07)	4.07 (0.89)	4.01 (1)	4.11 (0.92)

SA = Strongly agree, A = Agree, NE = Do not know, D = Disagree, SD = Strongly disagree. * *p* < 0.05.

**Table 2 insects-15-00368-t002:** Frequency of the response for each item and issue about perceptions related to honeybee conservation.

Perception of Honeybees	% Frequency of Responses in Likert Scale						Adjusted Mean Score (SD)			
	SA	A	NE	D	SD	Total	Boys	Girls	City	Countryside
(P1) Honeybees are the most aggressive insects	3.5	14.3	20.9	32.6	28.7	2.31 (1.14)	2.2 (1.14)	2.46 (1.13)	2.31 (1.17)	2.36 (1.04)
(P2) As soon as you see a honeybee you try to kill it	8.7	8.7	10.9	36.2	35.4	2.19 (1.25)	2.23 (1.27)	2.15 (1.24)	2.15 (1.24)	2.3 (1.23)
(P3) Other insects would live more peacefully if honeybees disappeared	1.3	4.8	37.7	29.4	26.8	2.24 (0.95)	2.24 (0.94)	2.24 (0.97)	2.24 (0.96)	2.18 (0.92)
(P4) Seeing a honeybee makes you feel threatened	10.8	25.5	15.6	27.7	20.3	2.79 (1.32)	2.55 (1.26) *	3.09 (1.34) *	2.82 (1.3)	2.7 (1.37)
(P5) If you find a beehive near your home, you should try to destroy it	10	16.6	21.4	25.3	26.6	2.58 (1.31)	2.73 (1.36)	2.42 (1.24)	2.6 (1.32)	2.52 (1.3)
(P6) Killing honeybees is right for public protection	2.2	3.9	12.1	26	55.8	1.71 (0.97)	1.7 (0.95)	1.71 (1.02)	1.7 (0.98)	1.73 (0.97)
(P7) If there were no more honeybees, it would be more pleasant to go to the countryside or go hiking	14.1	23.3	21.1	24.2	17.2	2.93 (1.31)	2.78 (1.25) *	3.13 (1.37) *	3.00 (1.32) *	2.60 (1.3) *
(P8) You would be willing to give a beekeeper permission to place hives in the countryside or near your home	5.2	16.6	27.9	22.3	27.9	2.49 (1.21)	2.64 (1.24) *	2.30 (1.15) *	2.47 (1.18)	2.42 (1.26)

SA = Strongly agree, A = Agree, NE = Do not know, D = Disagree, SD = Strongly disagree. * *p* < 0.05.

## Data Availability

All data generated or analyzed during this study are included in this published article. Raw data are available at SAAF, University of Palermo (Ed.5A-R004).

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
