# Peer review of "Less Known Is More Feared—A Survey of Children’s Knowledge of and Attitudes towards Honeybees"

_insects, 2024, doi:10.3390/insects15050368_

Round 1
Reviewer 1 Report
Comments and Suggestions for Authors
General comment: The paper by Leto et al. covers the correlation between knowledge and attitudes toward bees. Although the questionnaire itself is quite interesting, the background of this study seems lacking clear and unbiased literature support. While I agree that honeybees can play a significant role in maintaining plant diversity and the stability of various ecosystems, the decline in pollinators is mainly related to specialized pollinators, not general pollinators such as the honeybee. Moreover, in the paper the authors often use ‘bees’ and ‘honeybees’ interchangeably, which is misleading. By justifying their research with the decline in pollinator abundance when you clearly mean only the honeybee, the authors may be further spreading some unreasonable ideas. Please check for example Iwasaki and Hogendorn, 2021. https://doi.org/10.1016/j.cois.2021.05.005
Please reconsider the title!
See my detailed comments below. Please remember, that those are only suggestions and if you do not egree with them, please explain your point of view. Also, the sentences with which it is suggested to replace the original ones can be changed to better fit the text.
Specific comments:
1. Simple summary:
Line 12, 16, 17: ‘honeybees’, not ‘bees’
Line 11-12: Rephrase e.g. to: ‘This research focused on the relationship between knowledge and perception of honeybees, which provide essential ecosystem services, in 12-14-year-old children.’ Omit ‘for their conservation’- managed honeybees should not be the focus of conservation efforts
Line 12-14: Rephrase to: ‘Children can play a key role in biodiversity conservation as they pass on knowledge to their families, who, in turn can further disseminate it.’ And place at the beginning of the summary as it is an introduction sentence.
Line 14-15: Maybe rephrase to: ‘A survey was conducted with 231 students in Palermo (Sicily, Italy).’
2. Abstract:
Line 21-24: see my general comment. You should think of the importance of honeybees in supporting ecosystem stability, food safety, etc. You may argue that the honeybees are threatened by a lot of diseases, irresponsible use of pesticides, etc. but the phenomenon cannot be qualified as ‘global decline’. Please reconsider your justification for the research conducted here.
3. Introduction
I suggest that you try to justify the reason to conduct the questionnaire highlighting the possible correlation between knowledge and attitudes toward honeybees and/or the role of education in shaping attitudes toward honeybees. Lines 38-51 and 62-63 suggest that honeybees should be given a special focus in pollinator conservation, while it can be treated as farm animal. Maybe wild honeybees or local populations should/can be protected, but not honeybees as in general.
Lines 85-92 and 113-121: This can/should be the core of the Introduction. Other parts about honeybee conservation should be rephrased.
Lines 122-128: The purpose of the study is clear, but Line 125: Delete. 'How they see" sounds like perception/opinion, not knowledge
4. Materials and methods
This section itself is written very well. Please, make sure that you use ‘honeybees’, not ‘bees’ (line 143, but also check the whole text).
However, I have some remarks about the design of the questionnaire. First of all, you again use ‘bees’ for ‘honeybees’. Then, in the Table 1. I can see some questions that in my opinion cannot be evaluated as proof of knowledge:
C1- This is interesting, because honeybee’s mouthparts theroretically allow it to bite. Do honeybees bite humans? While I was bitten multiple times by Osmia bees, I do not know how it is in the case of honeybees and I am looking forward to the answer. I am not even sure if it is what you meant here, not to mention how confused a respondent can be.
C5 - I am not sure why you combined these 2 statements. What was your idea here? Honeybees use nectar, true. Do they hunt other life forms for food? I do not think they are predators, but do children know what to answer when one part is true and the second one is false/debatable? In logic when one of two sentences is true, the combined sentence is accepted as true. I would be confused if I were to fill this survey…
C7- How can you tell if they like to sting or not? This does not sound like a question to evaluate knowledge, rather opinion/perception…
C11- How can you tell if it is true?
C14- Do you mean (…) if there were no more bees (no more than now) or if there were no bees anymore (no bees at all)?
Also, could you describe how the mean scores were adjusted?
5. Results
Lines 200-205: I think you should present that data, if not in the main text body, at least as supplementary material.
What does 'world of bees' mean? Honeybee ecology? Biology? Please use more relevant term here (Line 205, 209, 218 and check throughout the text).
Line 218-219: Do you mean any particular question/questions? Or the knowledge in general? I think not general, because you mentioned no significant differences between females and males in lines 200-205, but I am confused...
I think you have to state clearly in the Table 1. which items are for honeybee ecology, which for importance as pollinators , etc. and show means (that you mention in the text) for those items
Lines 243-245: I think you should present that data, if not in the main text body, at least as supplementary material.
Section 3.3.: How is this 'negative attitude', not just 'attitude'? How was negative attitude expressed? On what scale? Because I do not get the Y axis in Figure 3.
6. Discussion
Line 291: How does this question apply to insect morphology?
Line 294-295: Are you sure? They do have the proper mouthparts, but tend to bite smaller intruders, not necessarily humans. Please check Papachristoforou et al., 2012 https://doi.org/10.1371/journal.pone.0047432
Comments on the Quality of English LanguageOverall, the text is easy to follow and reads very well in terms of language quality. My main concern here is the interchangeable use of bee/s and honeybee/s, which in this context is a severe mistake. Some sentences need rephrasing (see my specific comments) to improve the style. Also, awkward phrases need to be changed (listed in the comments), but there were not many of them.
Author Response
Dear Reviewer
Thank you for the time you have dedicated to our work and for the very useful advice and suggestions you have provided us. We agree with the proposed corrections and changes which we accept as described below in detail and as highlighted in the text (apart from small revisions which we have not highlighted for greater clarity). We have shortened the introduction and cited the very interesting articles proposed. We are aware that Apis mellifera can sometimes be confused with the generic word bees or honey bees. We have clarified this aspect better in the materials and methods and limitations. In fact, we agree with you that the attention to honey bees has been superior and has also sometimes obscured the importance of other pollinators. Our work aims to highlight how knowledge is necessary for the protection of bees, which children generally consider as honey bees but which they sometimes do not distinguish, at least in Italy, not even from wasps. This is the reason for some questions that perhaps might seem bizarre but whose rationale we explain in more detail (answers are underlined). The same text is also in the attached file in pdf for clarity.
- Materials and methods
C1- This is interesting, because honeybee’s mouthparts theroretically allow it to bite. Do honeybees bite humans? While I was bitten multiple times by Osmia bees, I do not know how it is in the case of honeybees and I am looking forward to the answer. I am not even sure if it is what you meant here, not to mention how confused a respondent can be.
Dear Reviewer,
perhaps in English the question may seem bizarre but the question was asked precisely because in Italy and particularly in Sicily people normally say that bees "bite" and do not use the term "sting". The question was specifically aimed at understanding whether it is just a figure of speech or whether kids really think that bees bite. We added some more explanation on this aspects.
C5 - I am not sure why you combined these 2 statements. What was your idea here? Honeybees use nectar, true. Do they hunt other life forms for food? I do not think they are predators, but do children know what to answer when one part is true and the second one is false/debatable? In logic when one of two sentences is true, the combined sentence is accepted as true. I would be confused if I were to fill this survey…
Dear Reviewer,
the children were asked the question whether bees, in addition to feeding on nectar, can hunt to obtain food. The questionnaire was designed so that a cross-response analysis could also be carried out. In any case, the contact person was always there with them to fill out the questionnaires and where the questions might seem unclear to someone it was possible to ask for clarification on the question.
C7- How can you tell if they like to sting or not? This does not sound like a question to evaluate knowledge, rather opinion/perception….
A honey bee that is away from the hive foraging for nectar or pollen will rarely sting, except when stepped on or roughly handled. Honey bees will actively seek out and sting when they perceive the hive to be threatened, often being alerted to this by the release of attack pheromones (below). The logic of the question was precisely to know if the kids know that bees sting when they perceive a threat to them or to the hive, therefore mainly for defense and not for attack. For this reason, it was included in the group of questions relating to knowledge.
C11- How can you tell if it is true?
Dear Reviewer,
Thank you for highlight this point. Our point comes from the following considerations.
In Italy, as in other countries, there have been many television reports and articles in newspapers, as well as educational broadcasts in which they highlighted the importance of bees for agriculture. The question aimed to understand what idea the children considered that certainly some of them had come into contact with this information offered to the general public.
C14- Do you mean (…) if there were no more bees (no more than now) or if there were no bees anymore (no bees at all)?
Dear Reviewer, the question cannot present any ambiguity in Italian, the language in which the questionnaires were submitted. It is obviously an almost "provocative" question in which we mean if there were no more bees (absolutely). The question was asked precisely to highlight whether the children had an idea of the great importance and role of these insects.
Also, could you describe how the mean scores were adjusted?
The mean scores were adjusted because they are scores calculated on the basis of categorical variables
Results
Lines 200-205: I think you should present that data, if not in the main text body, at least as supplementary material
Dear reviewer, thank you for the suggestion which we completely accept as it makes the text more complete. We have added the results of the statistical analysis in brackets and in fact we think that the work has acquired clarity and completeness.
What does 'world of bees' mean? Honeybee ecology? Biology? Please use more relevant term here (Line 205, 209, 218 and check throughout the text).
Dear Reviewer
we agree with you, in fact the expression "the world of bees" is redundant and can lead to confusion. We have deleted this expression in favor of the simple expression honeybees. ACCEPTED
Line 218-219: Do you mean any particular question/questions? Or the knowledge in general? I think not general, because you mentioned no significant differences between females and males in lines 200-205, but I am confused...
Dear Reviewer, thank you for bringing this point to our attention. Perhaps in the text we did not clearly highlight that they are two different analyses. The first (lines 200-205) was a chi-square calculated on the knowledge clusters, the second is a t-test calculated on the averages of the total score. We have clarified this aspect in the text.
I think you have to state clearly in the Table 1. which items are for honeybee ecology, which for importance as pollinators , etc. and show means (that you mention in the text) for those items
Dear reviewer, thank you for the suggestion, in fact the specific reference was missing and we have added a more specific reference to which item refers to the knowledge of honeybees ecology in the text.
Lines 243-245: I think you should present that data, if not in the main text body, at least as supplementary material.
Dear reviewer, thank you for the suggestion which we completely accept as it makes the text more complete. We have added the results of the statistical analysis in brackets and in fact we think that the work has acquired clarity and completeness.
Accepted
Section 3.3.: How is this 'negative attitude', not just 'attitude'? How was negative attitude expressed? On what scale? Because I do not get the Y axis in Figure 3.
Dear reviewer, this is a "negative attitude" because all the items are negatively oriented, therefore high scores correspond to a more negative perception towards bees. We have also included this clarification in the text.
- Discussion Line 291: How does this question apply to insect morphology? Accepted. We changed in basic knowledge of anatomy
Line 294-295: Are you sure? They do have the proper mouthparts, but tend to bite smaller intruders, not necessarily humans. Please check Papachristoforou et al., 2012 https://doi.org/10.1371/journal.pone.0047432.
ACCEPTED. Dear reviewer,
thanks for the very interesting suggestion. Our statement was more in reference to the fact that honeybees may bite as their primary action for feeding or defending themselves. Certainly, there are particular situations especially in relation to small invertebrates that are predators or that annoys bees invading or try to enter into the hive. The children of the age of those interviewed could not have been aware of this interesting discovery. However, we gladly accept the suggestion and have included this clarification in the text. In yellow we add this interesting suggestion, specifying that we assume that children do not know this specific behavior.

Reviewer 2 Report
Comments and Suggestions for Authors
Knowledge can save the honeybees!
Thank you for your paper. This research will be of interest for a wide range of audiences including teachers, researchers, and those working in conservation. While the research focuses on a small, localized population, its findings have relevance globally.
Please note the following parts of the paper that I feel need attention.
Overall: I see no major issues with the methodology, analyses or conclusions. The main problem I see with this paper have to do with the framing of the project. So, the issues are largely restricted to the introduction.
I have a major issue with the use of the phrases "ecological balance" and "ecological harmony" in the introduction. These concepts are common in the popular science literature, but are rarely invoked by ecologists. While I have not looked up every cited source, I would be surprised if any of those authors discuss "balance" and "harmony". These concepts need to be removed from the introduction. However, one of the survey questions uses the "ecological balance" wording. The use of the that phrase should be justified in the methodology.
The introduction: There are scholarship issues with the introduction. One example is in the sentence in lines 56-58 with the citation of source 11. Source 11 does not say what you are indicating in that sentence. In fact, you start the sentence with "various studies" but only cite source 11. Every source used in this introduction needs to be re-read and re-evaluated for its appropriateness.
The introduction is too long. The concepts are repeated. Careful editing is needed to make the introduction shorter and more concise.
line 40-41: "which might lead or be a con-cause of floral biodiversity decrement" makes no sense. Please correct.
Line 42: Remove the words "of pollinators" and insert the word "of" in that place
Line 53: Replace "is" with "are", add the word "well" after the word "most"
Line 54: What is "formation.... of the environment."? How are bees part of this process? More careful wording is needed here.
Line 78: I assume you want the word "honey' between "capita" and "consumption".
Lines 122-127: These lines need significant editing for clarity.
Materials and Methods.
Line 156: Why is Cronbach's alpha stuck here by itself? I think some explanation is justified. Is this statistic a method or a result? I can see why you consider it a method but, again, it's use and meaning need clarifying.
Paragraph lines 176-176. I don't understand how high scores indicate negative attitudes. "Scores... that indicated negative attitudes were: ... 5 points for "strongly disagree". " If you score 5 when you disagree with a negative statement, how, then are high scores indicative of negative attitudes? I think there is a wording problem in this paragraph. It should be corrected for clarity so other readers don't also get confused.
Discussion
Line 296: remove the last word "is"
Line 297: replace "seams" with "seems"
Comments on the Quality of English Language
My comments on language use are indicated in the full review. Editing for clarity is needed. Minor language problems need to be corrected.
Author Response
Dear Reviewer,
Thank you for your time and for the very useful advice and valuable suggestions which have certainly improved the work. As you suggested, we have streamlined the introduction and, in accordance with the suggestions of other reviewers, clarified some expressions. Below we report how we followed your instructions, all of which we willingly accepted. Please find in the file the specific answers to your requests that we all corrected and accepted.

Round 2
Reviewer 1 Report
Comments and Suggestions for Authors
I can see that the authors have considered some of the points I made. However, there are still some important aspects that need explanation. For the better clarity, please reconsider the previous and following comments:
1) First of all, the title does not give a clear idea of what the paper is about. You are stating that knowledge can save honeybees, whereas a) honeybees do not need to be “saved” as other bees do- you are right that there have been declines in the number of honeybee colonies locally but what you wrote in some points still suggest that honeybees should be an important focus of conservation actions, b) you do not show any impact that knowledge of students has on honeybee welfare. Your questionnaire itself is interesting, but the title in this context is not reasonable. Try to emphasize what the most important outcome of this research is or at least try to give a hint to a reader what this paper is really about. Maybe something like “Less known is more feared - a survey on children's knowledge of and attitudes toward honeybees”, as for what I understood, children with lesser knowledge showed more fear.
2) I cannot agree with your attitude to the interchangeable use of “bees” and “honeybees”. This is so confusing that you are stating that you used the term “bees” to ask about honeybees. Imagine that someone searched for papers dealing with the knowledge and attitudes toward bees in general and read your paper- he/she would feel misled, putting it mildly. If in the original questionnaire in Italian you asked about honeybees, just change it in the questionnaire so the translation is more accurate. If you do so, the additional explanations that you made are not necessary.
3) C7- I am still not convinced by your explanation. If you wanted to check if the children know that honeybees sting when in defense, and do not attack on purpose, why didn’t you just ask the question that way? Left in this form, the question is still about an opinion and do not prove the level of knowledge.
4) C11- I still do not consider this question as one about knowledge, rather than about opinion.
In general, please focus on the results of your questionnaire and main findings. Do not describe the links that you are not able to prove or did not intend to prove.
Also, the paper needs proofreading as the newer version contains more mistakes than the previous one. You can use one of some free online proofreading tools.
Comments on the Quality of English LanguageProofreading is necessary as in some places there are still grammar mistakes.
Author Response
Dear Reviewer,
Thank you for your time and for the very useful advice and valuable suggestions which have certainly improved the work. We accepted all your comments and adding some more clarifications (in yellow). We thank you very much for the proposed new title and we are glad to accept.
As you suggested, we have revised the English language clarifying also some expressions. Please find in the file the specific answers to your requests that we all corrected and accepted.
For the better clarity, please reconsider the previous and following comments:
1) First of all, the title does not give a clear idea of what the paper is about. You are stating that knowledge can save honeybees, whereas a) honeybees do not need to be “saved” as other bees do- you are right that there have been declines in the number of honeybee colonies locally but what you wrote in some points still suggest that honeybees should be an important focus of conservation actions, b) you do not show any impact that knowledge of students has on honeybee welfare. Your questionnaire itself is interesting, but the title in this context is not reasonable. Try to emphasize what the most important outcome of this research is or at least try to give a hint to a reader what this paper is really about. Maybe something like “Less known is more feared - a survey on children's knowledge of and attitudes toward honeybees”, as for what I understood, children with lesser knowledge showed more fear.
We accept the proposed Title
2) I cannot agree with your attitude to the interchangeable use of “bees” and “honeybees”. This is so confusing that you are stating that you used the term “bees” to ask about honeybees. Imagine that someone searched for papers dealing with the knowledge and attitudes toward bees in general and read your paper- he/she would feel misled, putting it mildly. If in the original questionnaire in Italian you asked about honeybees, just change it in the questionnaire so the translation is more accurate. If you do so, the additional explanations that you made are not necessary.
We accept to clarify and make a more appropriate translation with honeybees.
3) C7- I am still not convinced by your explanation. If you wanted to check if the children know that honeybees sting when in defense, and do not attack on purpose, why didn’t you just ask the question that way? Left in this form, the question is still about an opinion and do not prove the level of knowledge.
It might be is more a problem regarding the form; we are aware that it could have been expressed in a better way, however we chose to use jargon that is more easily understandable (or "closer") to the target; as regards the content, the item did not show problematic scores such that it had to be eliminated. In addition, Cronbach's alpha on the scale has an above-threshold value, which means that the content of the item is aligned with the content of others.
4) C11- I still do not consider this question as one about knowledge, rather than about opinion.
We do not agree. Regards C11 it seems clear to us that fewer bees would not benefit agriculture at all, therefore we consider it rightly categorized as knowledge and it could not be otherwise. To correctly answer this question, kids need to possess the notion that the role of bees is essential to pollinate agricultural crops, therefore it is knowledge. All of authors agree on this. The group of authors is composed by 1 professor in entomology, 1 beekeeper and Middle school teacher, and 1 associated professor in psychology, 1 researcher in psychology and education.
In general, please focus on the results of your questionnaire and main findings. Do not describe the links that you are not able to prove or did not intend to prove.
We reconsider some sentences.
Also, the paper needs proofreading as the newer version contains more mistakes than the previous one. You can use one of some free online proofreading tools.
We have revised the English thank you to an expert scientific English reviewer.
